# Transforming Agricultural Extension Service Delivery through Innovative Bottom–Up Climate-Resilient Agribusiness Farmer Field Schools

**Joab J. L. Osumba** [1,*] **, John W. Recha** [1] **and George W. Oroma** [2]

1   Consultative Group on International Agricultural Research (CGIAR) Research Program on Climate Change, Agriculture and Food Security in East Africa (CCAFS EA), International Livestock Research Institute (ILRI), Nairobi P.O. Box 30709-00100, Kenya; j.recha@cgiar.org
2   SNV Netherlands Development Organization, Uganda, Kampala P.O. Box 8339, Uganda; goroma@snv.org
*   Correspondence: J.Osumba@cgiar.org or jlosumba@gmail.com; Tel.: +254-722-408-387

**Abstract:** Conventional approaches to agricultural extension based on top–down technology transfer and information dissemination models are inadequate to help smallholder farmers tackle increasingly complex agroclimatic adversities. Innovative service delivery alternatives, such as field schools, exist but are mostly implemented in isolationistic silos with little effort to integrate them for cost reduction and greater technical effectiveness. This article presents a proof-of-concept effort to develop an innovative, climate-resilient field school methodology, integrating the attributes of Farmers' Field School, Climate Field School, Climate-Smart Agriculture and indigenous technical knowledge of weather indicators in one package to address the gaps, while sensitizing actors on implications for policy advocacy. Some 661 local facilitators, 32% of them women and 54% youth, were trained on the innovation across East Africa. The initiative has reached 36 agribusiness champions working with 237,250 smallholder farmers in Kenya, Tanzania and Uganda. Initial results show that the innovation is strengthening adaptation behaviour of agribusiness champions, farmers and supply chain actors, and reducing training costs. Preliminary findings indicate that the process is rapidly shaping group adaptive thinking. The integrated approach offers lessons to transform extension and to improve food security and resilience. The approach bundles the costs of previously separate processes into the cost of one joint, simultaneous process, while also strengthening technical service delivery through bundled messaging. Experience from this initiative can be leveraged to develop scalable participatory extension and training models, especially scaling out through farmer-to-farmer replication and scaling up through farmer group networks.

**Keywords:** integrated; participatory methodologies; policy; advocacy; agronomy; information; variability; agro-weather advisories

## 1. Introduction

Climate change and variability are adversely impacting agricultural production, food systems and food security in East Africa [1,2]. Climate change is projected to continue to impact overall crop yields negatively by as much as 5–72% in East Africa, averaging 24.3%, depending on the crop [3]. The need to increase agricultural productivity and improve agribusiness resilience against the backdrop of increased climate variability calls for adoption of more climate-resilient, more ecologically sustainable methods of agricultural production [3,4]. This call requires concerted investments by agricultural value actors and partners to support transformational change [5]. Actionable, sustainable climate information is critical for such a transformation [6–8]. However, currently, and especially due to the challenges of climatic "new normal" [9], most smallholder farmers do not receive actionable climate information for effective decision-making [2].

### 1.1. Agricultural Development, Agribusiness and Sustainability

Agricultural development entails transformation from subsistence or traditional modes of production to commercial, modern or scientific modes of agribusiness [10]. Depending on microclimatic and environmental conditions, this transformation often comes with social and ecological implications, especially degradation, emissions and pollution, and therefore requires rigorous sustainability safeguards [11,12]. Agribusiness is a blend of agriculture and business, combining economic activities or enterprises and actors or industrial outfits involved in an agricultural value chain from production, processing, storage, transportation and distribution; up to sales and marketing of agricultural products; to link producers to markets. In contrast, agricultural development is a process that creates conditions for the fulfilment of agricultural potential. Such conditions may include the generation of knowledge, dissemination of information, availability of technology and allocation of input. For sustainability, agricultural development needs to be as inclusive as possible, by using market-based approaches for the benefit of climate-vulnerable populations, stimulating sustainable agribusiness development, while delivering tangible impact and food security for the rural poor. Inclusive agribusiness is about the development of sustainable agribusiness solutions that expand the access to agricultural goods, services and livelihood opportunities for low-income communities in commercially viable ways. Through inclusion, key value chain players are assisted to identify economically viable food production and supply improvements and long-term solutions for increasing the reliability of food production and food supply. Through inclusive agribusiness, companies can create employment and other income-generating opportunities for the poor. They do this either directly or through value chains, such as suppliers, distributors, retailers or service providers. Examples of inclusive agribusiness practices include sourcing goods and services from smallholder farmers, facilitating access to financial services in rural areas, distributing and selling products in remote areas and supporting the development of local agro-enterprises.

There are many viewpoints on sustainability as a concept and on how it can be achieved, but generally the concept encompasses economic, social and ecological dimensions, focusing on strategies to meet the needs of the present generation without compromising the ability of future generations to meet their needs [13]. Sustainable approaches encourage economic businesses to frame decisions in terms of social and environmental considerations for the long-term. For instance, in climate change interventions, businesses are encouraged to pursue sustainability by sourcing products from climate-resilient clientele; using energy, water and other resources more efficiently; managing waste resourcefully and reducing greenhouse gas emission/carbon footprint. This sustainability challenge demands a response that integrates enhanced food security of the most vulnerable groups, with climate adaptation and mitigation of food crop production and supply systems, and a response that puts investment toward those interventions that lead to a market-driven uptake and scaling of inclusive climate smart business developments [5,14–17]. The integrated, participatory service delivery methodology presented in this paper is based on that sustainability principle [18].

### 1.2. Field School Approaches

A field school model, which is one of the delivery models in agricultural extension [19], is a group-based concept grounded on the principles of experiential adult learning. Field school approaches that incorporate and emphasize climate information can serve the purpose of climate-resilient agricultural development. A Farmer Field School (FFS) methodology is a bottom–up approach based on Robert Chambers' "farmer first" philosophy [20,21]. The methodology has been widely used to promote adaptation practices through social learning and capacity building [22–25]. FFS was initiated and driven by the Food and Agriculture Organization (FAO) through national ministries in charge of agriculture [26]. The FFS objective was to build common local knowledge, jointly with agronomists and users of the knowledge, for a more bottom–up, integrated production and pest management (IPPM) in a more sustainable way than the often top–down, agrochemical pesticide

approaches. According to FAO [26], "A Farmer Field School brings together a group of farmers, livestock herders or fisherfolk, to learn on how to shift toward more sustainable production practices, by better understanding complex agro-ecosystems and by enhancing ecosystem services." On the other hand, Climate Field School (CFS) methodology, another field school approach, was initiated and driven by the Global Framework for Climate Services (GFCS) Programme of the World Meteorological Organization (WMO), through the National Hydro-Meteorological Services (NHMS) [27,28]. CFS objectives were "to increase smallholder farmers' knowledge about climatological processes, to increase farmers' ability to anticipate extreme events in their agricultural planning, to improve farmers' capacities to observe climate variables, and to facilitate farmers' use of formal climate information in conjunction with their own experiences and knowledge in their management decisions" [29]. After following the program, farmers were expected to apply the climate information in setting up alternative crop management strategies [30]. Key features of similarity in the two approaches include season-long learning activities, learning or study/experimental plots to compare technologies and practices, facilitation to guide the learning and regular meetings/sessions during the season. Each session includes agroecosystem analysis (AESA) for FFS or agrometeorological analysis (AGROMETA) for CFS.

### 1.3. The Gap

CFS is promoted by WMO as good practice, akin to FAO's FFS Model, but the two are operated in silos separately from each other, hoping that practitioners can sequentially connect the dots and tie the knots [30]. In principle, CFS was patterned on the FFS concept, and looks similar to FFS on the surface, but in practice, the implementation did not live up to the FFS expectation [29–31]. Whereas FFS is practically cyclic and iterative [32], CFS is linear and unidirectional [31]. Running the two methodologies separately means they cannot be applicable concurrently for the same target group, because each has to run one step at a time. Moreover, the CFS approach strongly assumes that smallholder farmers are able to interpret scientific data or comprehend the data analytics and agro-weather advisories as disseminated. CFS promoters also give low priority to activities that build on farmers' local knowledge, capacities and institutional processes, a practice which represents a major departure from the original purpose of FFS. Moreover, CFS works in favor of conventional top–down models of extension service delivery, a style which creates barriers to optimization of the CFS–FFS synergy [32]. Instead of mirroring FFS, CFS ended up emphasizing more of dissemination, technology transfer and agro-weather advisories (i.e., prescriptions) than on-farm observation and knowledge co-creation. Currently, AESA is not emphasized in CFS, while AGROMETA is not emphasized in FFS. Further, integrating conventional with indigenous technical knowledge (ITK, which can be seen in the Appendix A) of weather prediction is not emphasized, both in CFS and in FFS [33]. Other key features of differences between Farmer Fields Schools and Climate Field Schools are presented in Table 1.

**Table 1.** Key features of differences between Farmer Fields Schools and Climate Field Schools.

| Factor | Principal Focus or Emphasis | |
|---|---|---|
| | **FFS** | **CFS** |
| Approach | Largely bottom–up [31,34,35] | Largely top–down [32] |
| Major focus | IPPM/AESA Experiments/PTD | Climate Analysis/AGROMETA, Weather advisories Demonstrations of "good practice" instead of "experimentation" to select the most locally suitable |
| Focal facility | Field site (e.g., farm) | Agrometeorological Station |
| Curriculum (Modules) | An agricultural commodity or resource e.g., plant, animal, soil, etc. | A meteorological hazard e.g., heat/cold stress, drought, flood, etc.; Translating technical terms to practical language [36] |
| Key strategy | Observation and knowledge co-generation | Dissemination, following the concept of technology transfer, focusing on how to use, not how to co-generate, climate information [7,28] |

Source: Author-constructed from the various sources cited in the table.

In an attempt to fill the gap, an innovative proof-of-concept comprising Climate-Smart Agriculture (CSA), FFS methodology, CFS and ITK modules has been proposed with these elements as a suitable combination [37]. Given the knowledge-intensive and multi-stakeholder nature of FFS, CFS, CSA and ITK, an innovative approach was needed to promote the harmonization of complementary attributes of these approaches. The innovative methodology integrates all four in one package, borrowing and embedding information and content from each to enrich the innovation. CGIAR Research Program on Climate Change, Agriculture and Food Security East Africa (CCAFS EA) is working with partners to make this innovation happen, by integrating climate resilience into the FFS to develop a climate-resilient agribusiness FFS (CRAFFS) approach for CSA. The initiative targets four categories of beneficiaries, namely (i) farmers and farmer organizations/cooperatives, (ii) all- and medium-sized enterprises (SMEs) in agribusiness, (iii) local service providers/extension agents and (iv) government officials/policy makers. Entry points include small scale agribusinesses (SMEs and farmer cooperatives), along with selected crop value chains [1], farming systems and institutional environment.

### 1.4. Objectives of the Study

The immediate objective of the intervention was to improve the decision-making skills of implementors in the CRAFFS approach, including the use of climate information to manage climate-related risks that prevent farmers from closing yield gaps. The medium-term objective was to improve agricultural productivity, build resilience and achieve climate change mitigation and co-benefits where possible. The ultimate objective was to increase the capacity of actors to apply climate-smart technologies, practices and innovations, with the aim of increasing their adoption among farmers, agribusiness SMEs and farmer cooperatives [38]. Specific objectives were to (i) equip trainees with knowledge about climate change, climate variability and climate-related risks affecting agriculture; (ii) provide participants with appropriate methodological tools to facilitate CRAFFS learning; (iii) prepare participants on how to plan CRAFFS implementation; (iv) prepare a climate-resilient crop production curriculum, with modules in the form of training aids for selected crops and (v) stimulate participants to share knowledge, skills and experience in local farming systems to improve production.

## 2. Materials and Methods

### 2.1. Study Area

The study area is the mandate area for Climate Resilient Agribusiness For Tomorrow (CRAFT) Project, comprising Kenya, Tanzania and Uganda [3,4]. Across the region, smallholder farmers face increased agro-weather risks, due to increased climate variability occasioned by climate change, manifesting in the form of more frequent and/or intense drought or prolonged dry spells, excessive rains/storms and/or floods, increased climate-induced pest and disease incidences and heat stress or frost waves, among others, with negative impacts on agricultural production through increased environmental degradation (soil, water, biodiversity and agroecosystems). Climate modelling studies indicate that temperature rise is affecting, and will continue to affect, rainfall patterns, both spatially and temporally, with significant adverse impacts on agricultural production, leading to risks of crop failure and food insecurity. Adverse weather conditions also directly affect agricultural marketing systems, leading to risks of market instability and food price volatility. It may also lead to disruptions in trade, supplies, sales and income.

### 2.2. Theory of Change/Impact Pathway/Results Chain

The theory of change or impact pathway constructed for this initiative was informed by, among others, hypothesized FFS results chains in [24,39]. A diagrammatic illustration is provided in Figure 1. Using the FFS approach, with additional climate information modules, the intervention focused on integrating climate-resilient agricultural practices in

the value chain development of selected crops from potato, cereals, pulses and oil crops in each of the three countries.

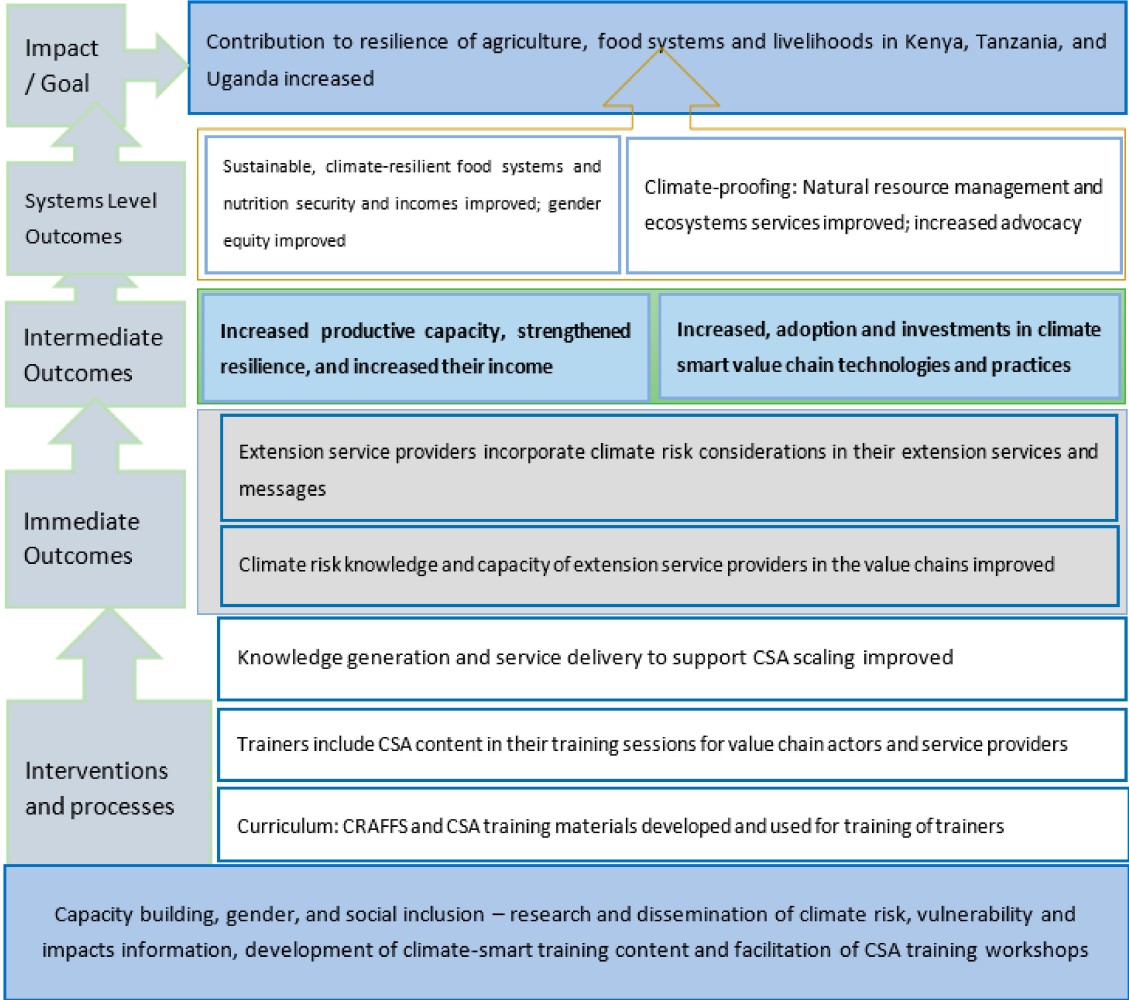

**Figure 1.** Theory of change/impact pathway for climate-resilient agribusiness Farmer Field School (CRAFFS) capacity building in Climate Resilient Agribusiness for Tomorrow (CRAFT). Source: authors.

### 2.3. Implementation

CRAFFS implementation in the three countries involved training of trainers or facilitators (ToT or ToF) and training of master trainers (MToT) of farmer leaders, farmer cooperatives, public and private agricultural extension officers, agribusiness SMEs, agro dealers and other service providers; initiating the CRAFFS through selected business cases coordinated by local facilitators; and expansion of the initiative to other groups in each country. Training duration was one week of 5 days for ToT/ToF and three weeks of 15 days for MToT, respectively. Given the CSA nature of the intervention, the entry point for the training was focused on climate-induced problem(s)/issue(s). Training events were led by FAO-trained FFS experts, with CCAFS EA experts providing input on climate change, climate trends, climate projections, weather forecast information and indigenous weather prediction. The training materials or modules were compiled and developed into a training manual, integrating FFS, CFS, CSA and ITK approaches [40].

The number of first-round trainees per country are presented in Table 2. Figure 2 shows images of training sessions held in Uganda in 2019. The basic CRAFFS learning cycle is presented in Figure 3. Participants were identified from partner business cases (agribusiness SMEs and farmer cooperatives), public and private agricultural extension agents, agro-

dealers and other local service providers [41]. The capacity building process involved employees of the partner SMEs and cooperatives, agribusiness project managers; agro-dealers; and their farmer representatives, plus subnational government agricultural officers and frontline, community-based extension agents, among others. Problem identification was based on local climatic experiences. Target enterprise for the training was based on value chains selected by partner business champions [41]. The training process comprised a bottom–up mixed methods approach of brainstorming, presentations, group work, plenary sessions and hands-on field practical. Brainstorming helped to ground the training on local conditions and circumstances. Presentations helped to provide snapshots of complex concepts. Groupwork helped participants to get acquainted with common adult learning and participatory rural appraisal (PRA) tools commonly used in FFS. Plenary sessions helped to sharpen facilitation skills and stimulate debates among participants. Field-based practical helped to bring the learning to real-world situations. Since the full training is season-long and COVID-19 interrupted the year 2020, the initial facilitators were allowed to continue working locally with the formed groups and will be graduated together with their farmer participants when they complete one learning cycle together.

**Table 2.** Trainees and business cases and targeted farmers per country.

| Country | Business Cases Lined up for Training in 2019 | Number of Participants Selected by the Business Cases in 2019 | Number of Farmers Targeted for Training in 2020 after 2019 ToT |
|---|---|---|---|
| Kenya | 11 | 107 | 23,200 |
| Tanzania | 08 | 215 | 24,500 |
| Uganda | 07 | 339 | 92,500 |
| Additional mobilization post-training | 10 | - | 97,050 |
| Total East Africa | 36 | 661 | 237,250 |

Source: Authors.

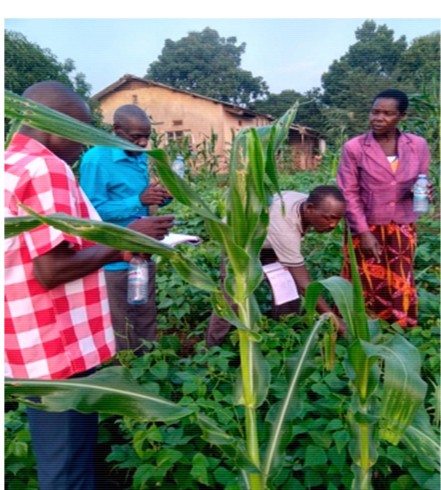

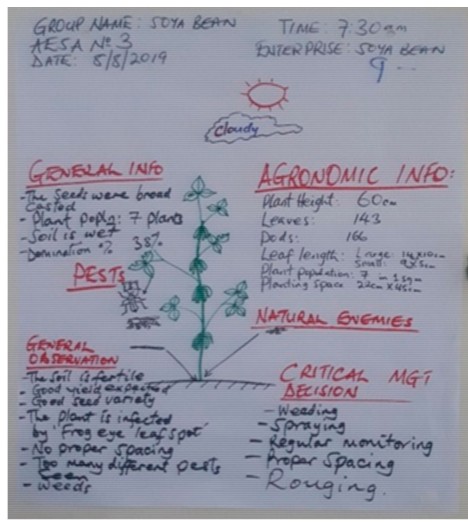

*Training of trainers (ToT) participants conducting an agroeco-system analysis (AESA) session in bean–maize intercrop in Lira, Uganda—July 2019*

*Soybeans drawing by ToT participants for AESA in Gulu—August 2019*

**Figure 2.** Images from training sessions in Uganda. Source: Courtesy of *Mr. Michael Ocircan p'Rajom, the FFS Trainer*.

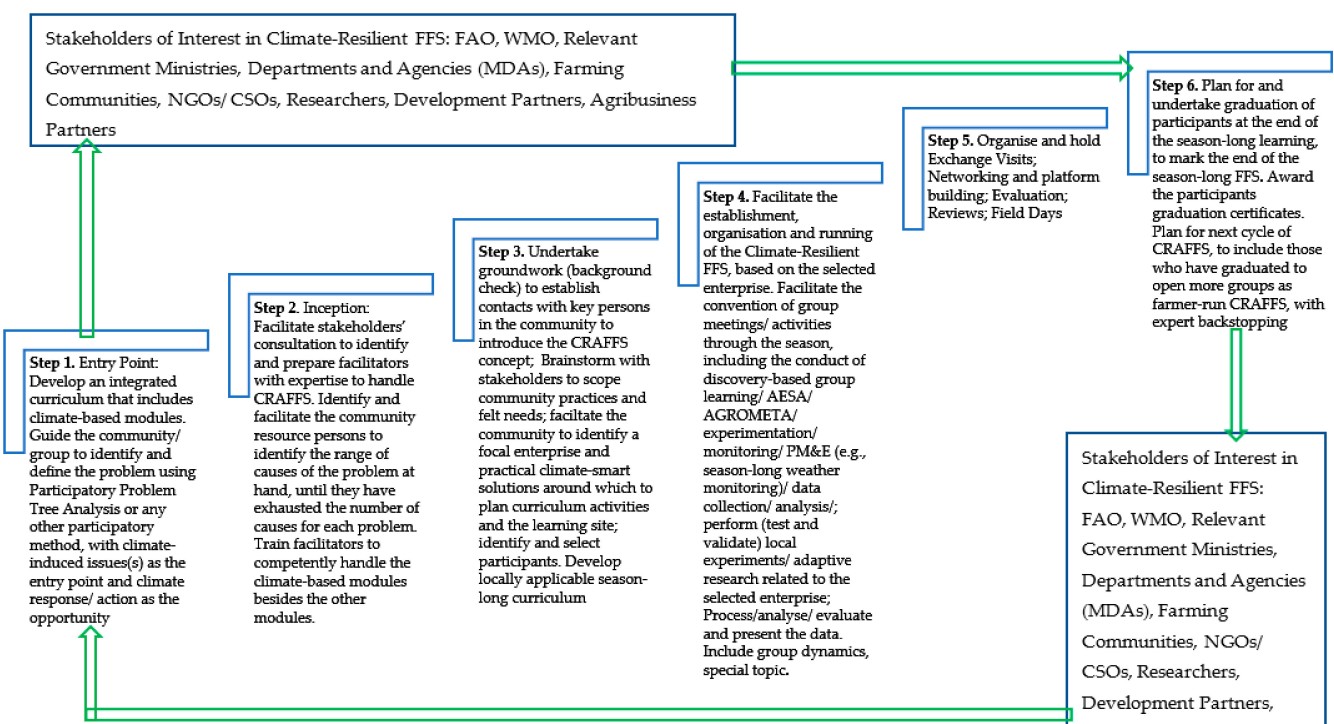

**Figure 3.** The basic CRAFFS season-long learning cycle. Source: synthesized and integrated from [29,35,36,40,42,43].

Candidates for the role of farmer–facilitator will be identified during the first CRAFFS graduation. The identified and selected farmer–facilitators will be taken for further training and be supported by an extensionist–facilitator to initiate and run a CRAFFS. At 32% women and 54% youth respectively, the training selection proactively addressed gender considerations and social inclusion by drawing more on women and youth. The process provides for a pretest and post-test that participants take at the beginning and end of training, to record how much they know at the beginning, how much they have learned from the process and how much they have gained from the education. The process also includes a semi-structured quiz and/or mood meters and a "most significant change" story method of capturing change, done at regular intervals. A provision is made in the climate change modules for crop–water–weather calendar monitoring and recording, to assist in AGROMETA in addition to AESA. Downscaled seasonal weather forecast information is provided to the FFS by the project modeling team and the local agro-meteorologist, before the CRAFFS team begins local seasonal monitoring for comparison. AGROMETA and AESA monitoring period is decided by the group, depending on the type and nature of the focal value chain.

Both conventional/scientific weather information and indigenous weather information (using agreed indicators) are observed, recorded, analyzed and reported. Data collection is done at predefined intervals using an integrated AESA/AGROMETA data sheet. Facilitators and participants reflect on evidence of key changes participants are observing, what indicates changes are occurring, how they are occurring and what is working or not working. Discussion was conducted to integrate both indigenous and conventional weather information results for better, more robust decision-making and appropriate action [40,44]. Storytelling is used as a way of communicating information and influencing change, but the storylines can also be used as a qualitative monitoring tool to track change.

### 2.4. Priority Value Chains Covered for CRAFFS Intervention

The CRAFFS ToT/ToF and MToT workshops were conducted for selected value chains in each of the CRAFT anchor countries. Priority value chains selected by business case champions and used for CRAFFS intervention trainings were common bean, green gram,

potato, sesame, sorghum, soybean and sunflower [45–56]. To support the process, the ToT/ToF/MToT and the project teams sensitized farmers and other value chain actors through raising awareness at institutional (policy) and local (community) levels. The effort included advocating for consideration of CRAFFS principles in national extension policies, strategies and funding mechanisms and developing the capacity of local partners/institutions to support CRAFFS and to partner with other organizations to create synergies. The priority value chains selected for intervention, namely potato, sorghum, common bean, green gram, soybean, sesame, and sunflower, are either those that are inherently climate-resilient but do not have organized supply chains and their value chains or markets are not yet well-developed, or those that their value chains and markets are relatively well-developed but need interventions in climate resilience or those that fall in both categories (Table 3).

**Table 3.** Priority value chains selected for intervention.

| Value Chain Cluster | Crop Value Chain | CSA Attributes of the Crop Value Chain | Market Supply Chain Attributes of the Value Chain | Focus of CSA Intervention Requiring CRAFFS |
|---|---|---|---|---|
| Roots and Tubers | Potato (target 9300 Smallholder Farmers—SHFs) | Sensitive to heat stress but does better than other crops in shorter rainfall regimes (due to shorter growing period and a higher water use efficiency) | Relatively well-developed, ready market in East Africa, e.g., vendors, hotels and restaurants; Potential for production improvement as staple and processed food | Contract farming with improved varieties |
| Cereals | Sorghum (target 24,000 SHFs) | More resilient to adverse climatic conditions than most cereal crops | Markets for high-quality grain for malting and food relief food in Kenya and Tanzania | Supply of agro-inputs targeting sorghum to increase yield from application of CSA technologies, practices and innovations |
| Pulses or Legumes | Common bean (target 6750 SHFs) | More Sensitive to heat stress than most other pulses but fixes nitrogen and can contribute reduction in external fertilizer application | Is one of the main agricultural commodities traded across East Africa: supply contracts with institutions, such as boarding schools, major hotels and restaurants | Common bean input and output trading with farmers, domestic and regional markets |
| | Green gram (target 10,700 SHFs) | Fixing nitrogen and can contribute reduction in external fertilizer application; is more climate-resilient than most other pulses | Demand from brokers/traders and supermarkets and institutional markets, and targeting to lock-in large buyers/processors | Providing access to climate-smart services and products to increase yield from CSA technologies, practices and innovations |
| | Soybean (target 49,500 SHFs) | It is more climate-resilient than other pulses; fixes nitrogen and can contribute to reduction in application of fertilizer | Demand for soybean is increasing in Uganda | Advocacy to include soybeans as a climate change adaptation strategy in national and local climate plans. |
| Oil Crops | Sesame (target 36,000 SHFs) | Drought tolerance and short growing cycle | One of the agricultural commodities traded across East Africa; export market is growing within the region | Promoting sesame with improved varieties and good agricultural practices; Advocating for sesame as an adaptation strategy |
| | Sunflower (target 43,500 SHFs) | Sensitive to temperature but fairly drought resistant | One of the commodities traded across East Africa. Its export market is growing | Adopt inclusive climate smart business technologies, practices and innovations |

Source: Synthesized from [41].

The main reasons for selecting the food crops were that the climate change projections and expected climate risks for the region are such that the food insecurity of many people in society will further aggravate; the cropping systems will be seriously affected by climate change; market developments for these crops show increasing consumption and sector growth; significant involvement of women and youth in production and supply of these food crops; growing private sector interests and a substantial investment potential; and possibilities to intercrop cereals with pulses and to rotate with other important crops.

*2.5. Agribusiness Partners and Farmers Reached with the CRAFFS Initiative under CRAFT*

The agribusiness partners or business case champions targeted by this initiative makes the intervention operate like a farmer business school, by taking the value chain approach to improve farm management and entrepreneurial decisions, based on contract farming [41,57]. By the close of December 2020, some 36 agribusiness partnerships had mobilized 237,250 farmers into CRAFFSs to implement CSA technologies, practices and innovations across Kenya, Tanzania and Uganda, despite the COVID-19 pandemic [58–62]. Improved decision-making emerges from an iterative process of analyzing the agreed indicators of agroecosystem health monitored through the season, considering the results from multiple viewpoints, making decisions accordingly, implementing the decisions and observing the new outcome [43]. The facilitator's role and duties include serving as catalyst, encouraging analysis, setting standards, posing questions and concerns, paying attention to group dynamics, serving as mediator and encouraging participants to ask questions and come to their own conclusions. The opportunity enables farmers to learn to improve their knowledge, change their attitudes and enhance their skills toward improved farm commercialization. Learning happens in the farm, but the curriculum covers the production cycle from planning to marketing, with practical exercises based on available resources. Actions proposed by agribusiness partners to achieve different CSA objectives, including synergies and tradeoffs, are presented in Table 4 [41]. Agribusiness objectives are combined with resilience objectives in the intervention to increase stability and sustainability, including the triple-win considerations for productivity, adaptation, mitigation and synergies where possible and tradeoffs where necessary. For synergies, some adaptation actions may end up achieving mitigation benefits and other co-benefits. Some mitigation actions may end up achieving adaptation benefits and other co-benefits. For tradeoffs, yield may be traded off for resilience in some situations, where necessary for stability of production.

**Table 4.** Climate-Smart Agriculture (CSA) aspects proposed/being implemented in the project funded business cases.

| CSA Pillar | Summary of the Interventions Business Case Champions Proposed for Implementation |
| --- | --- |
| Productivity | Improved, high-quality, high-yielding varieties; increased mechanization; enhanced soil testing and fertilizer use; increased refrigeration/preservation; market linkage for climate-smart products |
| Adaptation/resilience | Resilient, drought-tolerant/drought-escaping or excessive moisture-tolerant, pest- and disease-resistant, early maturing varieties; expanding agricultural land under CSA; conservation tillage/soil cover; enhanced water retention, storage and moisture conservation practices; intercrop diversification; drying facilities; Good Agricultural/Agronomic Practices (GAPs); cold chain/storage facilities; efficient irrigation technologies and practices; index-based crop insurance; access to better climate information and weather forecasting; improved, integrated pest and disease management; access to credit/financial services; grain storage facilities (e.g., hermetic bags/containers/silos, warehousing, etc.); diversified product chains; strengthening institutional frameworks/partnerships |
| Mitigation | Conservation tillage; sustainable ecological intensification; fertilizer use efficiency (emphasis: biofertilization); climate-smart mechanization; energy-efficient technologies (solar, refrigeration, processing, transport); sustainable soil management; reduction in postharvest losses; cost reduction |

Source: Synthesized from [41].

*2.6. Data Collection and Analysis*

(a) Data collection

Data was purposively collected from, about and through the 661 participants in the course of training and the 36 agribusinesses during the rollout in 2019 and 2020. Data collected included data from scoping studies, participation data (from awareness, sensitization and training attendance lists, disaggregated by gender and social strata), participant perception data (from Likert scale test scores), content of agribusiness proposals after the training (for content analysis and qualitative description of results), farmer/group targets and recruitment by agribusiness champions per value chain (for contract farming).

(b) Data analysis

Data analysis was done using Microsoft Excel and Statistical Packages for Social Sciences (SPSS) statistical packages for numerical or quantitative data. The parameters analyzed for training sessions include participant perceptions of the integrated methodology, relevance of the topics, topical coverage and method of delivery. The analysis also included the crops selected and business cases choosing them (by country, gender, age), and the farmers reached with the intervention.

(c) Qualitative data analysis (thematic analysis)

Qualitative data was generated from document reviews and content analysis of participant proposal documents and reports of capacity building activities and events, focus group reports or other text sources. This analysis focused on identifying and categorizing key themes (thematic analysis) to interpret patterns and meanings in the data, e.g., descriptions of participants' perceptions and experiences, interpreting patterns and narratives, interpreting the ideas and experiences of the participants and drawing conclusions from the narratives presented.

## 3. Results

Results presented in the form of qualitative descriptions, averages, frequencies and patterns.

*3.1. Training: Trainers and Master Trainers Trained by Country, Gender and Age Group*

In the first round of trainings in the year 2019, a total of 12 ToT/ToF sessions of about 50 individuals, each, were conducted for seven priority value chains. Some 661 local CRAFFS ToT/ToF were trained across the three countries, with an additional 76 MToT trained to backstop the ToT/ToF in subsequent steps of the process. Out of the local 661 ToT/ToF, 32% were women and 54% were youth (Figure 4). The lowest participant age was 20 years across the three countries, while the highest was 69, 65 and 72 for Kenya Tanzania and Uganda, respectively. The average age was 38, 37 and 34 for Kenya, Tanzania and Uganda, respectively.

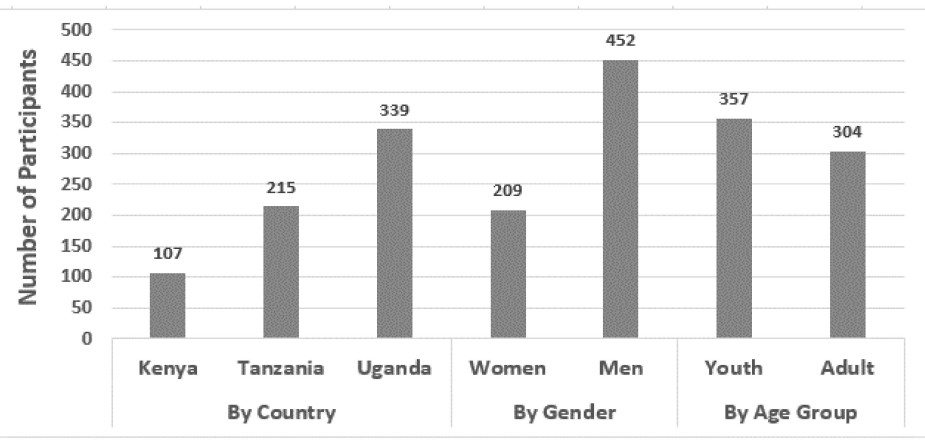

**Figure 4.** Facilitators trained by country and gender. Source: authors.

### 3.2. Pre- and Post-Test Results

Analysis of the daily training evaluation scores showed above-average satisfaction with the integrated CRAFFS approach, giving a score of 4.5 points on a scale of 5 points. The evaluation of the sessions was conducted using a Likert scale from strongly agreed, agreed and disagree to strongly disagree. Results of the pretest and post-test are presented in Figure 5. The results show that participant perception shifted greatly toward better satisfaction with what they gained during training

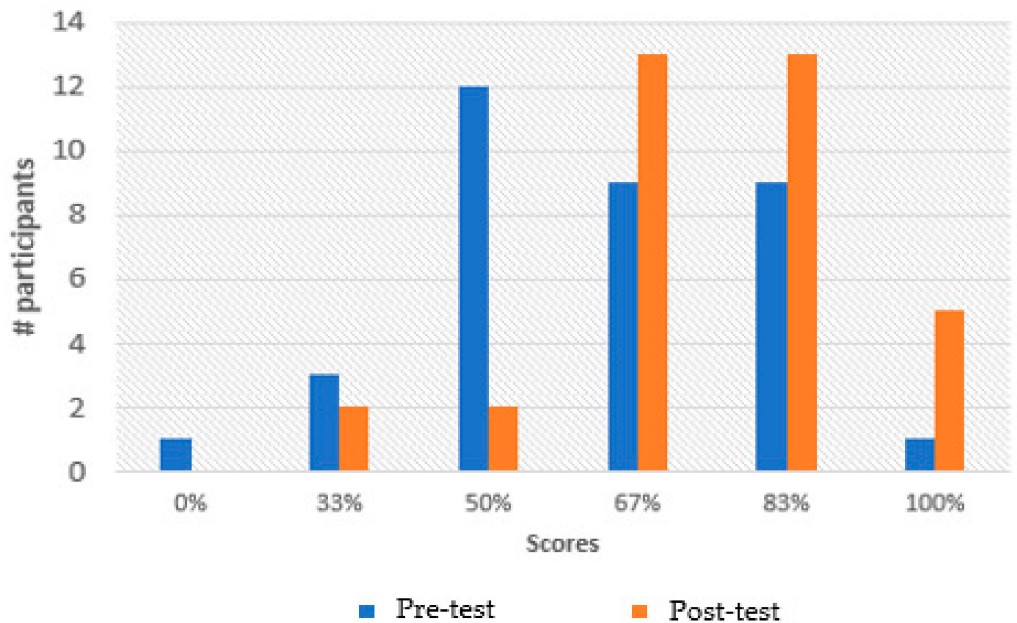

**Figure 5.** Learning evaluation of the pre- and post-test. Source: authors.

### 3.3. Business Cases and Farmers Reached by Country and Value Chain

Through the CRAFFS ToT/ToF and MToT implementation, by the end of 2020, the process had reached 29 business cases (Table 5), covering a total of 1004 farmers in Kenya, 16,247 farmers in Tanzania and 27,665 farmers in Uganda trained on CSA technologies, practices and innovations—a total of 44,916 (Figure 6). Additional mobilization of 97,050 farmers in 10 business cases post-training took the tally to 237,250 by end of 2020.

**Table 5.** Business cases reached by CRAFFS in Kenya, Tanzania and Uganda by December 2020.

| Country | Selected Crops by Business Cases by Country | | | | | | | Total |
|---|---|---|---|---|---|---|---|---|
| | Potato | Cereals | | Pulses | | Oil Crops | | |
| | | Sorghum | Green Grams | Common Bean | Soybean | Sesame | Sunflower | |
| Kenya | 2 | 3 | 2 | 1 | 0 | 0 | 0 | 8 |
| Tanzania | 2 | 2 | 0 | 2 | 0 | 0 | 8 | 14 |
| Uganda | 1 | 0 | 0 | 0 | 9 | 2 | 2 | 14 |
| Total businesses | 5 | 0 | 2 | 3 | 9 | 0 | 8 | 36 |
| Total farmers | 12,275 | 31,677 | 14,123 | 8909 | 65,334 | 47,516 | 57,415 | 237,250 |

Source: authors.

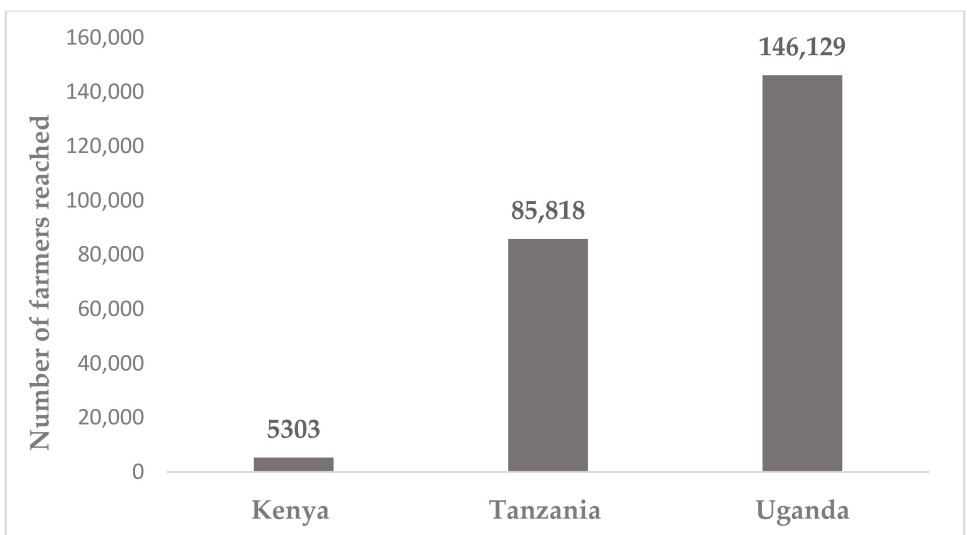

**Figure 6.** Number of farmers reached with CRAFFS interventions by country by December 2020. Source: Authors.

### 3.4. Main Findings and Key Points

(a) Summary of the principal components of the integrated methodology

A summary of the principal components of the integrated methodology is presented in Box 1, as one of the results (outputs) of the process.

(b) Effect of integrated CRAFFS on the motivation of participants

Early indications with CRAFFS and literature review findings on FFS show that it is possible and desirable to successfully adopt and apply an integrated approach to agroecosystem, agrometeorological and socioeconomic and agribusiness analysis. Pioneering cost-benefit studies show that field school approach is cost-effective, in terms of local capacity building per capita and participatory action research, as well as intermediate outcomes relating to knowledge, adoption, social capital and rural development [19,39,63–68]. However, the integrated, innovative approach is expected to be more cost-effective in the long run, because it bundles the costs of previously separate processes (FFS, CFS, ITK, CSA) into the cost of one process (CRAFFS). The findings suggest that methodological integration strengthened the impact of the information generated and the knowledge, skills and attitudes of the participants and other actors. Analysis of the pretest and post-test assessment indicated that the integrated CRAFFS training enriched participants' knowledge with climate information and climate-smart technologies, practices, and innovations. Participants mentioned the following elements of the course as the most useful idea captured during the training: CSA, climate change and weather forecast information; knowledge of CRAFFS; agronomic practices, including integrated pest and disease control; monitoring and evaluation of CRAFFS; presentation methods; agroecosystem analysis and organizing the farming calendar. Our experience with the capacity building and rollout of this integrated CRAFFS methodology shows that the integration has improved the relevance of the field school approach by simultaneously strengthening both agroecological literacy and agribusiness management with an appetite for agro-weather advisory information. Feedback from participants also indicated a strong need to bring both agronomists and agro-meteorologists to jointly collaborate from start, instead of one of them being the main agent and merely inviting the other, as currently happens in the traditional FFS and CFS.

**Box 1.** Summary of the principal components of the integrated methodology.

1. Conceptual background
   a. Introduction to CRAFFS
   b. From FFS to CFS: Climate-response farming—climate-related risks in agriculture, including risks of pests and diseases
   c. The CSA approach to develop agriculture under conditions of climate change
   d. Principles features of FFS, CFS, CSA and ITK, including ITK of weather and climate
   e. Group dynamics & teambuilding
2. ITK
   a. Indigenous knowledge/local wisdom
   b. Traditional weather prediction
   c. ITK indicators of climate monitoring and weather forecasting
   d. Gender and social inclusion analysis in climate smart ITK
3. FFS
   a. History and evolution of FFS
   b. The FFS process and FFS curriculum
   c. Agroecosystem analysis (AESA)
   d. Linking observed IPPM issues to their climate causes, and suggesting climate-related solutions or strategies to address the climate-related risks
   e. Governance and management in FFS and CRAFFS
4. CFS
   a. Basics of agro-weather monitoring (parameters and simple instruments), forecasting and interpretation
   b. Seasonal crop–water–weather calendar monitoring and crop–water relations management
   c. Pest and disease management based on weather information
   d. Agrometeorological instruments observation and innovation
   e. Agro-weather forecasts downscaling, and forecast products and advisories
   f. Cogeneration and applications of weather forecasts and use of climate information products (forecast interpretation, translation and communication)
   g. Agrometeorological analysis (AGROMETA): Observing and measuring weather parameters: weather instruments
5. CSA
   a. Agriculture, climate change
   b. Basics of crop microclimate and crop growth
   c. Climate change adaptation and strategies in agriculture
   d. Agricultural drought management
   e. Crop growth monitoring and analysis
   f. Climatic and ecological information for integrated crop–water–weather calendar
6. Climate-Smart Agribusiness
   a. Farming as a family business
   b. Agribusiness analysis
   c. Incorporating climate and weather forecast in decision-making
   d. Farm business management and agricultural marketing
   e. Incorporating climate forecast in decision-making
   f. AESA–AGROMETA Data Collection Sheet

Source: Synthesized from [40]

The trainings were instrumental in equipping participants with climate change and climate information knowledge; climate-informed agro-weather advisories and CSA knowledge, skills and attitude. The intervention was also used to sensitize agribusiness champions and value chain actors on what needs to be advocated on the policy front, like participation in the local development of downscaled climate information with meteorological agencies. Although the field school approach to extension service delivery is highly regarded in many rural development circles, the field school experience (in its various forms) has not been formally integrated into general, institutionalized service delivery

processes, especially in East Africa. Extension policy documents of Kenya, Tanzania and Uganda reviewed for this study mention field school methodology as one of the known extension approaches, but they all fall short of endorsing it as a preferred approach for extension service delivery. This is an area that requires further policy engagement with the governments. Kenya has noted it as an extension method in its national agricultural sector extension policy of 2012 but does not proceed to adopt it, in that document, as a method to promote in practice [69]. Document reviews for Tanzania show that farmer field school methodology is one the extension methods used in Tanzania, but there is no "one endorsed" approach by the government of Tanzania, although the national agriculture policy of Tanzania (of 2013) states that, "Junior Farmer Field and Life Schools (JFFLS) . . . shall be promoted" [70]. Uganda mentions it in its National Agricultural Extension Policy of 2016 and in the extension guidelines and standards of 2016 as one of the extension methods but does not expressly endorse it for promotion in the extension system [71].

(c) Key features and principles developed for the integrated CRAFFS

The key features and principles developed for CRAFFS are presented in Box 1. The purpose was to promote adaptive application of climate forecasts to crop farming decisions. The main objective of CRAFFS was to increase farmers' knowledge on the application of climate information in their decision making. Specific objectives are (1) to support the establishment of CRAFFS groups that have strong motivation to apply climate information as an input for making and taking climate informed agribusiness decisions, strategies, plans and actions and to protect the environment through climate change mitigation and (2) to strengthen understanding of weather, climate, climate variability, climate change, climate forecasts, climate monitoring and climate information applications; (3) participants then apply the climate information in practice through field-based pilot activities, in which they experiment with a range of planting techniques, varying approaches to integrated pest management and the utility of different seed varieties. (4) Participants learn how climate and weather information can be useful for operational purposes of farming. Farmers learn how to sufficiently downscale climate and weather information and how to interpret and apply the climate information for practical application in farming situations.

## 4. Discussion

### 4.1. The Innovation and the Context

As captured in Box 1 and 2, it is clear that technical innovation covered a wide range of content but placed climate risk management at the center of the messages [72]. The innovation showed that asking and responding to certain questions regarding indigenous indicators of weather monitoring can and do actually lead to determination of a weather forecast, just like the scientific methods of weather forecasting. In groups, participants were asked to share their experiences on local indigenous indicators of rainfall and to state what those indicators mean in the locality in terms of probability of weather events. As captured in the gender composition of the participants, age diversity, from 20 to 72 years, was very instrumental in gathering ITK, with gender and social diversity playing out well in the discussion, as the young learned from the old. The older generation had a wealth of experience in ITK, including experience of the climatic changes that have occurred over a long period of time. They were then asked to estimate how long it usually takes for that anticipated weather event to occur from the time the indicator manifests for the season. Based on the information generated from the foregoing questions, it was possible to hypothesize when the rains can be expected, and whether it is expected to be more or less or normal that season. Depending on the number of indigenous indicators the community identifies and the number of signs each indicator exhibits, it is possible to tally the observations and estimate percentages of the more, less or normal rainfall predictions (Appendix A). Armed with that information, farmers can begin their planning as they wait for the conventional weather forecast, which often comes later, to confirm or disprove the ITK estimates. Studies show when conventional weather forecasts are sufficiently downscaled to the local level, the precision level is very high between the indigenous, traditional, local estimates and

that of the conventional meteorological weather forecasts [73]. The training and simulation during training showed that combining indigenous weather forecasting with conventional weather forecasting improves reliability of weather forecasts and increases the confidence of users to depend on them.

**Box 2.** Key features and principles of the integrated CRAFFS.

CRAFFS is intended to facilitate participatory intervention activities, which can lead to the development of technologies suited to local conditions and preferences and adaptable to extreme and changing climate conditions, thereby increasing resilience in the face of climate change. Over and above the FFS/CFS principles [25,74], intervention deliberations generated the following principles to guide CRAFFS:

1. Identify problems based on farmers felt needs and interests but with climate change and climate risk as the reference point.
2. Climate information and climate-informed participation, including crop–water–weather calendar, should be at the center of all deliberations.
3. As the learners lead the studies, they should lead all studies, including simple weather trend monitoring and projection, recording and analysis. The discovery-based learning should include discovery and prediction of weather behavior from their own monitoring, recording and analysis.
4. The integrated curriculum should include content on climate and agro-weather aspects at all stages and phases. CRAFFS curriculum should to effectively incorporate climate literacy in the process. Agronomists should work more closely with available agrometeorological service providers to ensure that farmers are properly guided to participate appropriately in the "cogeneration" and application of climate information and climate-informed "agro-weather" advisories for their agricultural production purposes. They should jointly collaborate more closely during CRAFFS facilitation, to ensure that farmers are properly guided to participate appropriately in the cogeneration and application of climate information and agro-weather advisories for their agricultural production purposes. They should effectively collaborate from the start, instead of one of them being the main agent and merely inviting the other, as currently happens in the traditional FFS and CFS.
5. Apart from the seasonal and natural cycle of the practice and the experiments being investigated, the training should also follow, monitor and analyze the climatic cycle that goes with the season. The season should begin with a participatory agro-weather scenario planning and end with a participatory agro-weather review.
6. Learning materials should include simple gadgets for local monitoring of weather and climate and should always be consistent with local conditions, be less expensive to develop and be controlled by the learners, themselves. The role of facilitators or subject matter specialists, including agro-meteorologists, should mainly be to provide backstopping support on interpreting the results.
   a. The hands-on learning by doing should include simple hands-on discovery for weather observations, monitoring, recording and analysis
   b. Farmers' local knowledge—alongside science-based knowledge—coproduces and cocreates new knowledge, science and public services.
7. Field school group members should be given an opportunity to observe and monitor local weather and climate and to make site-specific records with the backstopping support from technical facilitators. The learning field site should provide opportunities both for agroecosystem observations and for simple agrometeorological observations, including through traditional, indigenous indicators of weather prediction. Host team subgroups should collect and analyze weather data in the field in the period between meeting days, suggest action decisions based on the analyses of the data and present the results and suggestions to the other group members in the field school for discussion/debate, questioning and refinement.
   a. The indigenous and local knowledge, used within the FFS learning process as an important source of information, should include indigenous and local knowledge of climate and weather prediction, including local indicators of weather prediction.
   b. The group diversity should have members with experience in indigenous weather monitoring indicators.
8. Agrometeorological analysis should be conducted through local simulation just as with agroecosystem analysis, including the linkages between agro-weather trends and agroecosystem observations, with the experts facilitating instead of presenting.
9. Group meetings are to begin with host team reports and analysis of weather trends for the intervening period between meetings during the season, and weather projection for the next intervening period up to the next meeting; recommendations and informed decisions by the group for the next intervening period includes plans for subsequent activities that recognize reported weather projections.
10. The skills and competencies targeted for participants should include skills and competencies in climate and weather information and knowledge management.
11. Conclusions and implementation should consider results of climate weather monitoring and analysis to enhance decision-making.
Source: Synthesized from [40]

The process component of the innovation is about the design and purpose of the field school. FFS was first conceptualized to respond to the problem of pest resistance in Asia, so

the design was made to respond to integrated pest management or IPM [75,76]. However, when the concept was scaled to Africa, the major challenge in Africa was productivity and not pests, so in Africa, FFS was first designed to respond to the productivity challenge. Therefore, productivity was added to IPM as a priority for Sub-Saharan Africa, hence the IPPM [34]. Currently, climate is major threat worldwide, and therefore, FFS needs to be resigned to respond to the challenge of climate risk management, i.e., FFS for CRM. This is the process innovation that this article conceptualizes, to combine ITK, FFS, CFS and CSA under one conceptual framework to tackle climate change, while reducing the cost of doing so [40].

Literature search for this paper showed that there is a dearth of rigorous scientific studies on FFS and/or CFS, especially in peer-reviewed journals [39,66]. Much of what has been studied is about impact and less on innovation [64,77,78]. The innovation described in this article is an attempt to contribute to that need. What is available is mostly grey literature evaluations, reviews and assessments captured in technical notes, reports and case studies and operating manuals [79]. Literature review of peer-reviewed journal articles available on the topic point out that the field school methodology, and its outcomes or impacts, is still largely uncritiqued, and that most of the studies reviewed "had weaknesses in reporting on sampling, analysis, and presentation of data" [39,68]. However, despite continuing discourse around issues of outcomes, such as cost-effectiveness, information sharing and communication, scalability, sustainability, participation and financing and technical impacts, systematic reviews also identify field school methodology as a promising model, in terms of capacity building and technology adoption [19,34,66,79,80]. Globally, reviews of agricultural (farmer, agropastoral, agribusiness, etc.) field school initiatives show that the approach has become a model for agricultural/agropastoral education in many parts of the world [75,80]. The integrated CRAFFS model presented here has been lauded by participants, project staff and project managers as technically more comprehensive and financially more cost-effective than each of them implemented individually and separately, as has been the case previously [81,82]. CRAFFS comes in as an alternative approach to enhance uptake and adoption of technologies, especially under conditions of climate change [38]. Integrating FFS, CFS and CSA in one participatory methodology package innovatively reduces costs, while strengthening the climate resilience of agroecosystems and agricultural livelihoods of the rural poor, especially women and youth, and enhances the adaptive capacity, with mitigation co-benefits among the agribusiness beneficiaries and their value chain actors [42].

### 4.2. The Relevance of the Integrated CRAFFS Methodology

The principal purpose of promoting climate-resilient participatory methodologies, such as CRAFFS, is to institutionalize adaptive application of climate information and agro-weather forecasts to crop farming decisions, adapting their farming practices throughout the season [40,80,83–86]. Much of the methodological strategy and content for the integrated CRAFFS model focuses on the reduction of climate risk in agricultural production. It is also meant to train growers on the measurement of rainfall and the observation of weather and climate implications for fields and crops in a standardized way as the basis of a CRAFFS. This involves adapting the choice of crops, crop varieties, planting dates and other cultural measures, while, at the same time, managing and manipulating the soil, water and microclimate, where possible. The idea is based on the "abstract-to-concrete continuum," which asserts that learning becomes more meaningful when abstract learning and concrete experiences are related and combined [87], and that learners do retain and recall only 20% of what they hear but retain and recall 30% of what they see, 50% of what they hear and see, 70% of what they say and discuss, 80% of what they do and experience and 90–95% of what they do and explain to others [81,82,87,88]. The integration strategy is to support actions that build more resilient agricultural production systems, increase institutional capacities to use the CRAFFS as a platform for raising awareness and introducing adaptation into farming practices and to advocate for climate change response policies and strategies. The innovation helps to deepen the strengthening of farmers' capacities to analyze their production systems, identify local problems and test possible solutions

and, eventually, encourages them to adopt and adapt practices most suitable to their local farming systems. It focuses on group learning by observation, discovery and experimentation and validation in comparison plots (as opposed to demonstration in model farms). It brings together concepts and methods from agroecology, agroclimatology and experiential learning, through regular field studies, group discussion and analysis of results, exchange of experiences and informed, collective decision-making. The trainings were also used to sensitize agribusiness case champions and agricultural value chain actors on what needs to be demanded on the policy front, like the provision of downscaled climate information services from meteorological agencies [40]. The training sensitized participants to demand the downscaling of climate information services to the localities of the participants for relevance in decision making at the local level.

### 4.3. Conceptual Framework: Integrated CRAFFS for Institutional and Policy Engagement

Like formal, localized agricultural research initiatives, agriculture-based field school tools and methods focus on identifying concrete solutions for local problems, but they apply different styles of experimentation and analysis [43]. However, both of them build local capacity for critical analysis and practical decision-making on how to manage local ecosystems, and both stimulate local innovation, while emphasizing principles and processes, rather than recipes or technology packages. Globally, reviews of agricultural (farmer, agropastoral, agribusiness, etc.) field school initiatives show that the approach has become a model for agricultural/agropastoral education in many parts of the world [34,75,80]. This integrated CRAFFS model strengthens that adult learning and the building of local capacity. However, to effectively incorporate climate literacy in the process, participants expressed strong opinions on the need for agronomists to work more closely with available agrometeorological service providers to ensure that farmers are properly guided to participate appropriately in the "cogeneration" and application of climate information and climate-informed "agro-weather" advisories for their agricultural production purposes. The kind of institutional framework that reflects the feedback from participants is presented in Figure 7. The field schools being formed will be coalesced into a movement of CSA CRAFFS networks to pursue this advocacy agenda from the ground [29,89–91]. The proposal in Figure 7 will require significant institutional commitment and support, which is currently being offered by the CRAFT project but will need institutional sustainability, driven by the private sector, when CRAFT folds up.

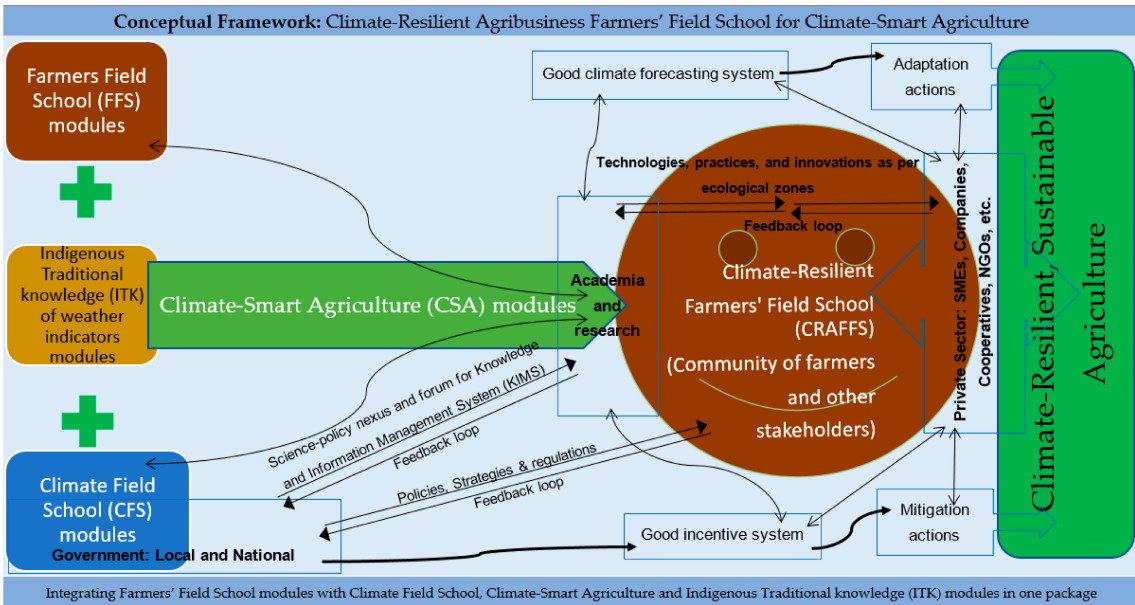

**Figure 7.** Conceptual framework for the innovative, integrated, Climate-Resilient Agribusiness Farmers' Field School for Climate-Smart Agriculture. Source(s): authors.

## 5. Conclusions and Recommendations

This article has presented lessons on the conceptualization and application of an innovative, integrated, climate-resilient agribusiness farmer field school process for climate-smart agriculture in East Africa. The article presents an effort to develop a methodology for integrating the attributes of FFS, CFS, CSA and ITK of local weather indicators in one package to address the gaps and to reduce costs. The integrated approach is expected to be more cost-effective in the long run, because it bundles the costs of previously separate processes into the cost of one process, while also strengthening technical service delivery.

The intervention aimed at enhancing both technical and process innovation to transform service delivery, drawing insights from the FFS CFS, CSA and ITK approaches to develop the CRAFFS methodology, combining the use of sustainable production practices with climate information [22]. The approach integrates FFS and CFS, instead of treating them separately, as is the case in current practice. CRAFFS emphasizes both AESA and AGROMETA equitably, as opposed to the current separate FFS and CFS approaches, each of which emphasizes its own AESA or AGROMETA, respectively. The article has elucidated implementation experiences or outputs, technical aspects, organizational issues, institutional and policy implications of an innovative, integrated CRAFFS methodology. The authors' experience with the capacity building and rollout of this CRAFFS methodology shows that the integration has strengthened the relevance of the field school approach by simultaneously strengthening both agroecological literacy and agroclimatic sensitization with increased appetite for agro-weather information and advisories to improve agribusiness management. The integrated methodology proved to be an eye-opener for the trainers (experts), the trainees (participants), project staff and project management, because they were coming into contact with this integration for the first time. A handbook [40] has been developed for the methodology. Agribusiness champions adopted the innovation by including aspects of the integrated methodology in their agribusiness proposals and by leveraging funds for implementation of the proposed innovative actions.

The major takeaway from the study is that stakeholders should engage policy makers to get their opinion on formal support for the methodology in formal agricultural extension systems, especially the innovative, integrated version conceptualized and presented in this article, which promises to be more cost-effective than its different components, when they are handled separately in isolation. This may be achieved by integration and complementing CRAFFS with other modes of extension, dissemination and communication, while maintaining its core principles. The lessons from this experience can be leveraged to create scalable participatory extension and training models throughout the region, especially through farmer-to-farmer social marketing and replication methods by observation techniques and scaling up through farmer group networks, if relevant authorities can develop an enabling environment and sustainability plan for CRAFFS. What will be needed is how best to support the scaling-up and the institutionalization of the methodology and the policy dialogue necessary to bring about institutionalization. For sustainability of the innovation, policies promoting participatory approaches, replacing policies promoting off-the-shelf technologies and input packages, may also be necessary. Given that the integration is at its proof-of-concept stages, further trials and interrogation may still be needed to strengthen the evidence of its positive attributes.

Finally, lessons from the COVID-19 pandemic also call for the need to explore the possibility of developing digital climate-oriented farmers' field schools that can operate despite pandemics, using mobile information and communication technologies (ICT) opportunities [59]. Further, FAO has provided guidelines on how to conduct CRAFFS under of COVID-19 rules [58,60]. A good example of digital FFS is documented in [92]. CRAFFS groups can use apps and prioritize digital solutions to set up informal networks for information sharing.

**Author Contributions:** Conceptualization: J.J.L.O. and G.W.O.; Methodology: J.J.L.O. and G.W.O.; Validation: J.J.L.O. and G.W.O.; Formal analysis: J.J.L.O.; Investigation: J.J.L.O. and J.W.R.; Data

curation: G.W.O.; Writing—original draft preparation: J.J.L.O.; Writing—review and editing: J.W.R.; Visualization: J.J.L.O.; Supervision: J.W.R.; Project administration: J.W.R. and G.W.O. All authors have read and agreed to the published version of the manuscript.

**Funding:** This research was funded by the Netherlands Ministry of Foreign Affairs (DGIS) through grant reference Climate Smart Agriculture in East Africa; Application no. 4000000819. The funds are administered by the SNV Netherlands Development Organisation to implement CRAFT in Kenya, Tanzania and Uganda. This research was also in part financially supported through CCAFS grant reference D7540 for Accelerating Impacts of CGIAR Climate Research for Africa (AICCRA)—ESA regional project.

**Institutional Review Board Statement:** The study was conducted according to the guidelines of the Declaration of Helsinki, and approved by the Institutional Review Board (or Ethics Committee) of the International Livestock Research Institute (ILRI), on 20 December 2018.

**Informed Consent Statement:** Informed consent was obtained from all subjects involved in the study in Kenya, Tanzania and Uganda.

**Data Availability Statement:** The data presented in this study are available on request from the corresponding author. The data are not publicly available due to confidentiality requirements of the donor for the project.

**Acknowledgments:** The authors acknowledge CRAFT colleagues including notable support and contribution from Teferi Demissie and Helena Shilomboleni (CCAFS EA), James Kebirungi (SNV Uganda), Oscar Nzoka and Joyce Mbingo (SNV Kenya) and Godfrey Kabuka and Emmanuel Nkenja (SNV Tanzania). The CRAFT project (2018–2023) is funded by the Ministry of Foreign Affairs of the Netherlands. The CRAFT project is implemented by SNV (lead) in partnership with Wageningen University and Research (WUR); CGIAR's Research Program on Climate Change; Agriculture and Food Security (CCAFS); Agriterra and Rabo Partnerships in Kenya, Tanzania and Uganda.

**Conflicts of Interest:** The authors declare no conflict of interest. The study was conducted within a project context, within the mandate of the project.

## Appendix A

**Table A1.** Indigenous technical knowledge (ITK) local weather indicator monitoring and tracking tool.

| Indigenous Indicator of Weather in Your Area | Local Meaning for Weather in the Locality (in Terms of Whether There Will Be Rain or Drought, or Whether the Rains Will Be Normal, More or Less) | How Long Does It Take for the Prediction after Witnessing the Indicator | Have You Already Seen It This Season? | In View of, and Based on, the Information so far Available from the Weather Indicator, by the Rating of this Locality: | | What Farming Plans or Actions or Strategies will you Adopt Based on This Traditional Weather Forecast Information Available to You? |
|---|---|---|---|---|---|---|
| | | | | Are the Rains for the Season Likely to be (1) Normal, (2) above Normal or (3) below Normal? | When (Dates Range) is the Rain likely to Start if It Will Come? | |
| Indicator 1: | | | | | | |
| Indicator 2: | | | | | | |
| Indicator 3: | | | | | | |
| Indicator etc . . . | | | | | | |
| as many as communities can identify | | | | | | |
| Total tally for normal, above normal and below normal scores | Summary of meanings for the indicators | Summary of lead times for indicator prediction | Tallies for Yes and No | Normal = X% Above normal = Y% Below normal = Z% | Summary of expected dates of onset | Summary of adaptation options suggested |

Source: authors.

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
