# Peer review of "Transforming Agricultural Extension Service Delivery through Innovative Bottom–Up Climate-Resilient Agribusiness Farmer Field Schools"

_sustainability, doi:10.3390/su13073938_

Round 1
Reviewer 1 Report
In the manuscript, the authors presented an important point, although the manuscript has some drawbacks:
1. Intruduction - in the introduction, please write more about sustainability. The manuscript was sent to Sustainability. This is missing.
Please write more about agribusiness, agricultural development. Suggested publications:
- Roman, M.; Roman, M.; Roman, K. Spatial differentiation of particulates emission resulting from agricultural production in Poland. Agricultural Economics Zemědělská ekonomika 2019, 65, 375-384. https://doi.org/10.17221/337/2018-AGRICECON
- Ayaz, M.; Feizienė, D.; Tilvikienė, V.; Akhtar, K.; Stulpinaitė, U.; Iqbal, R. Biochar Role in the Sustainability of Agriculture and Environment. Sustainability 2021, 13, 1330. https://doi.org/10.3390/su13031330
3. Conclusions - in your conclusions, please also answer the following question: what is new to this manuscript?
Author Response
Detailed response in the uploaded file

Reviewer 2 Report
The main objective of this paper is to argue that a blended approach of the FFS and the CFS could help smallholder farmers in east Africa to improve agricultural practices in the situation of climate change related adversities. The paper seems to have originality in this field in terms of topic, methodology, and arguments. However, it is recommended the paper, particularly chapter 3, “Results”, should be revised to show main findings more clearly. It is quite hard to draw directly key lessons from the empirical results. Five discussions in chapter 4, which of headings are wrong numbered, also are not precisely connected to the findings. So the reviewer suggests the authors to revise the paper for readers to distinctly understand the connection of arguments with the findings, and if possible, to add a concise conceptual framework of the paper other than a lot of figures, tables, and photos.
Author Response
Detailed response in the uploaded file

Round 2
Reviewer 1 Report
Missing from the manuscript I proposed to write more about agricultural development:
- Roman, M.; Roman, M.; Roman, K. Spatial differentiation of particulates emission resulting from agricultural production in Poland. Agricultural Economics Zemědělská ekonomika 2019, 65, 375-384. https://doi.org/10.17221/337/2018-AGRICECON
- Ayaz, M.; Feizienė, D.; Tilvikienė, V.; Akhtar, K.; Stulpinaitė, U.; Iqbal, R. Biochar Role in the Sustainability of Agriculture and Environment. Sustainability 2021, 13, 1330. https://doi.org/10.3390/su13031330
Author Response
Detailed response in the uploaded file

Reviewer 2 Report
Despite major revision, the length of the manuscript should be reduced with a focus of main findings and arguments.
Author Response
Detailed response in the uploaded file
